# Efficient Maternal to Neonate Transfer of Neutralizing Antibodies after SARS-CoV-2 Vaccination with BNT162b2: A Case-Report and Discussion of the Literature

**DOI:** 10.3390/vaccines9080907

**Published:** 2021-08-15

**Authors:** Jonathan Douxfils, Constant Gillot, Émilie De Gottal, Stéphanie Vandervinne, Jean-Louis Bayart, Jean-Michel Dogné, Julien Favresse

**Affiliations:** 1Qualiblood s.a., 5000 Namur, Belgium; 2Namur Thrombosis and Hemostasis Center, Namur Research Institute for Life Sciences, Department of Pharmacy, Faculty of Medicine, University of Namur, 5000 Namur, Belgium; constant.gillot@unamur.be (C.G.); jean-michel.dogne@unamur.be (J.-M.D.); j.favresse@labstluc.be (J.F.); 3Département de Gynécologie, Centre Hospitalier Régional de Huy, 4500 Liège, Belgium; emilie.degottal@gmail.com; 4Laboratoire de Biologie Clinique, Centre Hospitalier Régional Huy, 4500 Liège, Belgium; stevdv25@gmail.com; 5Department of Laboratory Medicine, Clinique Saint-Pierre Ottignies, 1340 Ottignies, Belgium; jean-louis.bayart@cspo.be; 6Department of Laboratory Medicine, Clinique Saint-Luc Bouge, 5004 Namur, Belgium

**Keywords:** neonates, infant, COVID-19, vaccine, antibodies, pregnancy, immunity

## Abstract

This case reports on the successful maternal to fetal transfer of neutralizing antibodies after vaccination with BNT162b2 in a pregnant woman at 25 weeks of gestation. The levels of neutralizing antibodies were approximately 5-fold higher in the umbilical cord than in the maternal blood while the level of total antibodies showed only a 2-fold increase. This suggest that the antibodies that crossed the syncytiotrophoblast cell barrier have specific characteristics that correlate to functional neutralizing capacity. Although pregnant and lactating women have been excluded from clinical trials for several reasons including ethical concerns about fetal exposure, accumulating evidence has now revealed that these vaccines are safe and efficient for both the fetus and the woman. Vaccination against COVID-19 in pregnancy is vital to control disease burden and to decrease morbidity in the ante-, peri- and post-natal periods. Inclusion of pregnant women in research programs for the development of SARS-CoV-2 vaccines should be mandatory to provide this population with the equitable benefits of vaccine research.

## 1. Introduction

Before the vaccine rollout, multiple cohort studies documented that pregnant woman were more susceptible to severe COVID-19 compared to age-matched non-pregnant women [1,2,3]. This is in line with the initial expectations, which stated that according to immunologic and cardiopulmonary adaptations occurring during pregnancy, the risk of severe illness from respiratory infections typically increases [4]. A systematic review of 60 studies on SARS-CoV-2 in pregnancy reported that severe illness occurred in up to 18% of pregnant patients and critical disease complicated up to 5% of the cases, comparable to rates observed in the general population [5]. Nevertheless, despite recommendations from public health advocates for pregnant women including the Center for Disease Control and Prevention (CDC), the American College of Obstetricians and Gynecologists (ACOG), and the American Academy of Pediatrics, pregnant or breastfeeding women have been excluded from clinical trials during the development of existing COVID-19 vaccines [6,7,8,9,10,11,12]. This causes large gaps in understanding the safety and efficacy of these vaccines in this population, which can be definitively detrimental to vaccine acceptance. Although data are emerging documenting the efficacy and the safety of these vaccines [13] and although guidance from these data recommend vaccination in pregnant and breastfeeding women, [14,15,16] getting vaccinated during pregnancy mainly remains a personal choice. Thus, the acceptance of COVID-19 vaccination in pregnancy is an important concern. According to a survey study, the COVID-19 vaccine acceptance rate during pregnancy was 58.4%, a rate which is consistent with the acceptance rate of other recommended vaccines in pregnancy but that is not sufficient to avoid deleterious cases of neonate infection due to maternal to fetal transmission or infection during the early stages of life [17,18].

Both neonates and pregnant women are particularly susceptible to respiratory infections, including influenza and respiratory syncytial virus (RSV) [19,20,21], and recent data demonstrate that a greater proportion of neonates and infants have severe or critical illness upon SARS-CoV-2 infection compared to older children [22,23,24]. Rare cases of vertical transmission have been reported in neonates [5], but histopathological changes in the placenta have suggested that the inflammatory nature of SARS-CoV-2 infection during pregnancy could cause adverse obstetric and neonatal events [25,26]. The risks of SARS-CoV-2 infection during pregnancy are preterm delivery, preeclampsia, emergency cesarian delivery, and neonatal complications including respiratory distress or pneumonia, disseminated intravascular coagulation, asphyxia, and perinatal deaths [27,28]. In addition, placental injury may increase the risk of long-term adverse neurodevelopmental, as is the case for other conditions that affect the placenta through the release of proinflammatory cytokines such as interleukin-6, which leads to the activation of T helper cell 17 [29].

Neonates rely on the transfer of maternal immunoglobulin G (IgG) across the placenta for protection against pathogens, and vaccination aims to mimic the presence of pathogens to stimulate the maternal immune response. Data from the literature show that maternal to neonatal transfer of anti-SARS-CoV-2 antibodies is efficient although the maternal to fetal transfer ratio of IgG antibodies may differ depending on gestational age and the type of immunization, i.e., disease- or vaccine-induced [30,31,32,33]. There is also a significant improvement in the transfer of spike specific IgG into the umbilical cord with the time from second dose, suggesting that time from vaccination may be an important determinant of the transfer rates of specific IgG subpopulations following immunization in pregnancy [32]. Despite reassuring data, the neutralizing capacity of these transferred antibodies was not assessed [30,32,33].

This case-report assessed the neutralizing capacity of umbilical cord blood compared to maternal blood in addition to conventional serological assay in a woman who received ta two-dose regimen of BNT162b2 during the antenatal period. To appreciate the magnitude of the antibody response of this dyad, a comparison with an age-matched cohort will also be done.

## 2. Case Description and Methods

A 28-year-old pregnant woman received her first dose of BNT162b2 mRNA COVID-19 vaccine (Pfizer-BioNTech, Mainz, Germany) at 25 weeks of gestation. The second was administered at 30 weeks. No problem was reported during the antenatal period, and no severe adverse events were reported after the vaccination. The patient only complained headache and tiredness the day after the second dose, which resolved with the administration of paracetamol twice a day. The delivery was planned to be on the 4th of July but occurred 8 days earlier, on the 26th of June at the Centre Hospitalier Régional (CHR) of Huy (Liège, Belgium), 90 days after the administration of the first vaccine dose. No delivery complications were reported, and all of the parameters of the newborn were normal. At 4 weeks, the newborn was still doing well. On the day of delivery, umbilical cord blood was collected as well as maternal blood to permit comparison of the level of SARS-CoV-2 neutralizing antibodies.

To compare the results obtained with the umbilical cord and maternal blood samples, 13 non-pregnant women of a similar age (i.e., 25 to 35 years of age) with no history of previous SARS-CoV-2 infection (i.e., no documented positive RT-PCR and the absence of anti-nucleocapsid antibodies), were included. These vaccinated controls were recruited at the Clinique Saint-Luc Bouge (Namur, Belgium) due to their participation in the CRO-VAX HCP study, a study which has already been described in detail elsewhere [34,35,36]. All of the participants provided informed consent prior to specimen and data collection. The study was approved by a central ethical committee (EudraCT registration number: 2020-006149-21).

Blood samples were collected in SST™ II advanced tubes (BD Vacutainer, Plymouth, UK) and were processed according to the manufacturer recommendations to obtain serum. Serum samples were then aliquoted and were stored at −20 °C until analysis. Antibodies against the SARS-CoV-2 nucleocapsid (anti-NCP; Elecsys Anti-SARS-CoV-2 NCP total qualitative ECLIA, Roche Diagnostics, Machelen, Belgium) and the receptor binding domain of the S1 subunit of the spike protein (anti-S; Elecsys anti-SARS-CoV-2 spike total quantitative ECLIA, Roche Diagnostics) were measured at each time point. Results above 0.8 U/mL (manufacturer’s cut-off) or 0.165 COI (cut-off index as found previously [37]) for anti-S and anti-NCP antibodies, respectively, were considered positives.

Neutralizing capacity was estimated by performing a surrogate virus neutralization test (sVNT) [38]. The iFlash-2019-nCoV NAbs assay (performed on an iFlash1800 analyzer from Shenzhen YHLO Biotech Co., Ltd. (Shenzhen, China)) is a one-step competitive paramagnetic particle chemiluminescent immunoassay (CLIA) for the quantitative determination of 2019-nCoV NAbs in human serum and plasma. The assay detects NAbs that block the binding of RBD and ACE2. First, NAbs (if present) react with the RBD antigen coated on paramagnetic microparticles to form a complex. Second, the acridinium-ester-labeled ACE2 conjugate is added to competitively bind to the RBD-coated particles that were not neutralized by the NAbs (if present) from the sample, and these form another reaction mixture. Under a magnetic field, magnetic particles are adsorbed to the wall of the reaction tube, and unbound materials are washed away by the wash buffer. The resulting chemiluminescent reaction is measured in relative light units (RLUs), with an inverse relationship between the amount of NAbs and the RLU value being detected. A result <10.0 arbitrary unit (AU)/mL is considered negative, and a result >10.0 AU/mL is considered positive (according to the manufacturer’s information). The neutralizing capacity was also confirmed using a pseudovirus neutralization test, as previously described [38]. Samples are considered to be neutralizing if the dilution factor provides a 50% diminution of the relative infectivity that is superior to 1:20. The agreement of the sVNT assay with the pVNT was found to be 97.2% [38].

## 3. Results and Discussion

The total anti-RBD titers of the maternal and the umbilical cord blood were 1120 U/mL and 2349 U/mL compared to 1747 U/mL (geometric mean; 95%CI, 1231–2438 U/mL) in the control group at the identical time point, i.e., 90 days post-vaccination. The NAbs titers were 163 and 906 AU/mL for the maternal and the umbilical cord blood, respectively, compared to 427 AU/mL (geometric mean; 95%CI, 222–820 AU/mL) for the control group at 90 days post-vaccination. The maternal to fetal transfer ratio was 2.10 for total anti-RBD antibodies and 5.56 for NAbs. A summary of the results is presented in Table 1 and Figure 1. The neutralizing capacity of the maternal and umbilical cord blood was confirmed by the pVNT, with both samples having higher dilution factors than 1:20 to half of the relative infectivity.

Recent investigations suggested that maternal SARS-CoV-2 immunization, either induced by the disease or by the vaccination, may provide neonatal protection through the transplacental transfer of antibodies [30,31,32,33]. Of particular importance was the demonstration that antibody transfer is correlated with the time from vaccination to delivery, which may allow future determination of the optimal timing of COVID-19 vaccination in pregnant women [33]. Compared to the non-pregnant controls, pregnant women exhibit quite comparable vaccine-specific antibody titers, [39] although IgG titers have also been found to be slightly lower in pregnant women [13].

Zdanowski et al. observed a slightly lower maternal to fetal ratio, i.e., 1.28 ± 0.80 [33]. Mittal et al. also report that the mean IgG titer is similar in maternal and infant sera, but the average gestational age at first vaccine dose was 33 weeks [40]. In our case, 13 weeks separate the first vaccine dose to delivery, which could then explain the higher ratio observed. This observation was also made by Beharier et al. and Mittal et al., and the same phenomenon has been also reported after SARS-CoV-2 infection [30,40]. Thus, the data in the literature are consistent and should now permit us to draw hypothesis on when to vaccine pregnant women in order to achieve the best maternal to fetal transfer ratio. A first BNT162b2 dose administered during the second trimester could permit high umbilical cord neutralizing antibody titers that are even higher than the ones observed in the maternal blood to be obtained (Table 1).

Interestingly, our case also confirms the presence of neutralizing antibodies in the umbilical cord blood (Table 1). Another case demonstrated that the antibody subtypes, which are transferred, possess a neutralizing capacity, although a clear description of the neutralization assay that was used was not provided [41]. It is also interesting to note that total antibodies and NAbs titers in the maternal blood were at the lower range in our control cohort (Figure 1). This may be due to individual factors or may be the consequence of the maternal to fetal transfer of IgG antibodies, reducing the level of antibodies in the maternal blood.

The observations that we made in this case report following vaccination with BNT162b2 are not so surprising. For most pathogens, umbilical cord titers of IgG are higher than in maternal blood due to endosomal transport of IgG across the syncytiotrophoblast cell barrier from maternal to fetal circulation [42,43,44]. Among these IgG antibodies, galactosylated IgG1 antibodies are transferred preferentially, followed by IgG3, IgG2, and IgG4 [45]. This is potentially the result of enhanced binding to both placental FcRN and FCGR3, enabling the selective transfer of specific antibody subpopulations to most effectively arm neonates in the setting of pathogen exposure [45,46]. The fact that the ratio for neutralizing antibodies is higher than the one of total antibodies may also be explained by this preferential transfer of IgG1, as most serological assays used to assess the humoral response towards SARS-CoV-2 vaccination target either all types of antibodies or all IgG subtypes. Indeed, it is possible that specific Fc glycoforms can be linked to Fab specificity and subsequently to a different neutralizing capacity, as reported for influenza vaccination [47]. Thus, if a specific fraction of neutralizing galactosylated IgG subtypes is transferred from the maternal blood to the fetus, the neutralizing capacity ratio may increase more sharply than the total level of antibodies since other types of antibodies, which are not neutralizing and do not transfer trough the placenta, can compensate for the decrease of neutralizing antibodies in the maternal blood.

Interestingly, in line with the above explanations, immunization transfer may also depend on whether it is acquired after vaccination or previous infection. Indeed, some cases were reported and described inefficient maternal to fetal transfer following COVID-19 disease during pregnancy [48,49]. This could have been explained by Rh isoimmunization, which can compete for IgG transfer within the neonatal Fc receptor (FcRN), but, despite high levels of anti-D antibodies, protective antibodies for other pathogens (i.e., rubeola and varicella) successfully crossed the placenta [48]. This observation can be explained, however, by changes to the Fc-glycosylation profile of the SARS-CoV-2 antibodies due to the inflammatory state generated by SARS-CoV-2 infection [31,50].

This may influence the neutralizing capacity of the antibodies and may impact the transfer of immunity, especially during the third trimester [31,47,50]. Nonetheless, this perturbed placental transfer is probably dependent on the trimester, as observed across two independent third trimester-infection cohorts where normalized following second trimester infection, suggesting that inflammation-induced alterations in SARS-CoV-2-specific-glycan profiles may resolve over time from infection [31]. It remains to be known if such alterations in Fc glycosylation are also observed after vaccination, but in any case, our results and those from other groups [30,32,33] strongly suggest that BNT162b2 administration in the second trimester permit the generation of a strong immune response that can be efficiently transferred to the fetus in order to protect neonates from SARS-CoV-2 infection.

Although these data have been obtained in a small number of subjects, these are quite reassuring, and the present case demonstrates that the antibody subtypes that are transferred possess high neutralizing capacity.

## 4. Conclusions

This case reports on the successful maternal to fetal transfer of neutralizing antibodies after vaccination with BNT162b2 in a pregnant woman at 25 weeks of gestation. The levels of neutralizing antibodies were approximately 5-fold higher in the umbilical cord than in the maternal blood, while the level of total antibodies showed only a 2-fold increase. This suggests that the antibodies that crossed the syncytiotrophoblast cell barrier have specific characteristics that correlate to functional neutralizing capacity. Although pregnant and lactating woman have been excluded from clinical trials for several reasons including ethical concerns about fetal exposure, accumulating evidence has now revealed that these vaccines are safe and efficient for both the fetus and the woman. Vaccination against COVID-19 in pregnancy is vital to control disease burden and to decrease morbidity in the ante-, peri- and post-natal periods. Inclusion of pregnant women in research programs for the development of SARS-CoV-2 vaccines should be mandatory to provide this population with the equitable benefits of vaccine research.

## Figures and Tables

**Figure 1 vaccines-09-00907-f001:**
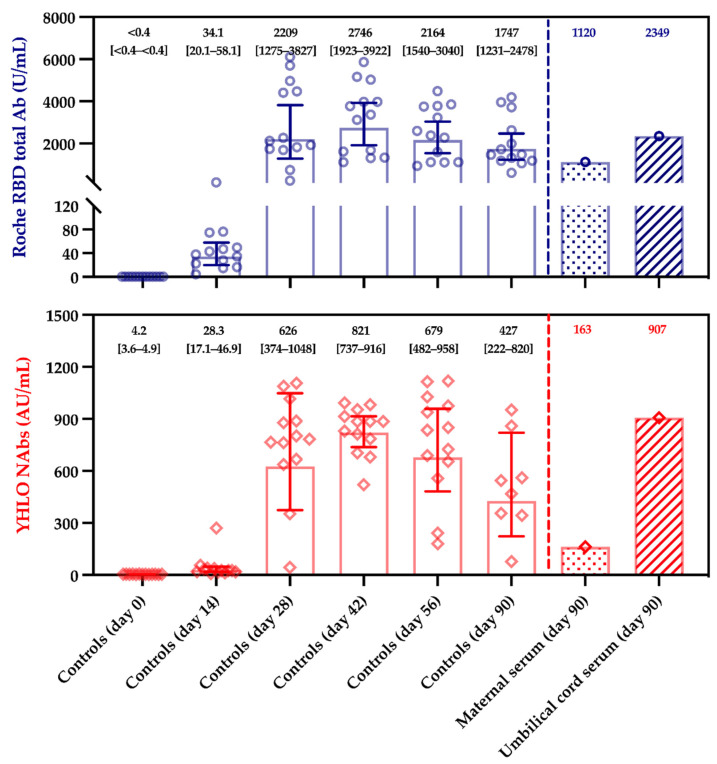
Serological response in controls, maternal and umbilical cord sera. Control sera selected from the CRO-VAX HCP study cohort and included 13 non-pregnant women aged from 25 to 35 (median, 30 years of age; min–max range, 25–35), without history of SARS-CoV-2 infection. Blood was collected at baseline (day 0), day 14, day 28, day 42, day 56, and day 90. Maternal and umbilical cord blood were only collected on day 90 after the vaccination, at the time of delivery. Positivity thresholds for the Roche RBD total Ab and the YHLO NAbs assays were 0.8 U/mL and 10.0 AU/mL, respectively.

**Table 1 vaccines-09-00907-t001:** Summary of serological results obtained from the maternal and the umbilical cord blood samples. For pVNT, the highest the dilution factor, the stronger the neutralizing capacity.

Serum Samples	Anti-N Titer (U/mL)	Anti-S Titer (U/mL)	sVNT (AU/mL)	pVNT (Dilution Factor)
Maternal	<0.165 (negative)	1120	162.9	1/40 (positive)
Umbilical cord blood	<0.165 (negative)	2349	906.4	1/60 (positive)
Maternal to fetal transfer ratio ^†^	/	2.10	5.56	1.50 ^‡^

^†^ Ratio is defined as the results of umbilical cord blood divided by maternal blood. ^‡^ pVNT is not a calibrated quantitative method, and thus, the ratio is only given for information. Abbreviations: pVNT, pseudovirus neutralization test; N, nucleocapsid; NR, non-reactive; S, spike; sVNT, surrogate virus neutralization test.

## Data Availability

The data presented in this study are available upon request from the corresponding author. The data are not publicly available due to ethical and privacy reasons.

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
