# Peer review of "Efficient Maternal to Neonate Transfer of Neutralizing Antibodies after SARS-CoV-2 Vaccination with BNT162b2: A Case-Report and Discussion of the Literature"

_vaccines, 2021, doi:10.3390/vaccines9080907_

Round 1
Reviewer 1 Report
The case report by Douxfils and colleagues describes the total and surrogate neutralizing antibodies from maternal and umbilical cord blood from a pregnant women vaccinated with BNT162b2 RNA vaccine against SARS-CoV-2. While it is now known that immunization of pregnant women does not harm the mother or fetus and confirms previous work on vaccination of pregnant women with the BNT162b2 RNA vaccine, the timing of the vaccine during pregnancy still needs to be determined. The findings in this manuscript may provide new information in this regard. I have several minor comments regrading the manuscript.
1) Lines 15, 17, 164, 179, 221, 223: In this case report the authors use a surrogate virus neutralization assay that measures antibodies to receptor binding domain of SARS-CoV-2 spike protein. Since this is not a classical virus neutralization test, the authors should state that it is a “surrogate virus neutralization test.”
2) Line 25, 232: recommend changing “vaccines is mandatory” to “vaccines should be mandatory.”
3) line 124: please define “AU.”
4) Lines 119, 129, 133, 168, 205: Please change “Nabs” to “sNAbs.”
5) Line 155-157: Here the authors discuss the ratios of maternal to fetal antibodies and draw conclusions based on their one sample. It is difficult to draw a conclusion based on an n= 1. Thus explanation should be remove and conclusions included into the next sentence.
6) Sentence on lines 167-169: This sentence should be rewritten. Try, “It is also interesting to note that total antibodies and sNAbs titers in the maternal blood were at the lower range of our control cohort (Figure 1).”
Author Response
The case report by Douxfils and colleagues describes the total and surrogate neutralizing antibodies from maternal and umbilical cord blood from a pregnant women vaccinated with BNT162b2 RNA vaccine against SARS-CoV-2. While it is now known that immunization of pregnant women does not harm the mother or fetus and confirms previous work on vaccination of pregnant women with the BNT162b2 RNA vaccine, the timing of the vaccine during pregnancy still needs to be determined. The findings in this manuscript may provide new information in this regard. I have several minor comments regrading the manuscript.
- Lines 15, 17, 164, 179, 221, 223: In this case report the authors use a surrogate virus neutralization assay that measures antibodies to receptor binding domain of SARS-CoV-2 spike protein. Since this is not a classical virus neutralization test, the authors should state that it is a “surrogate virus neutralization test.”
We thank the reviewer for this remark. To avoid confusion for the readership, and as this technique is stated in the material and methods as being a surrogate virus neutralization test, we decided to keep the term neutralizing antibodies. Nevertheless, the reviewer correctly pointed out this limitation and we therefore confirm the neutralizing capacity using a pseudovirus neutralization test, as previously described.[1] This is now stated in the material and method section as well as in the result section. Thus, we add:
“The neutralizing capacity was also confirmed using a pseudovirus neutralization test, as previously described.[1] Samples are considered as neutralizing if the dilution factor to provide a 50% diminution of the relative infectivity is superior to 1:20. The agreement of the sVNT assay with the pVNT has been found to be 97.2%.[1]”
and
“The neutralizing capacity of the maternal and umbilical cord blood has been confirmed by the pVNT, both samples having higher dilution factors than 1:20 to half the relative infectivity.”
Results have also been added into Table 1.
- Line 25, 232: recommend changing “vaccines is mandatory” to “vaccines should be mandatory.”
This has been changed accordingly.
- line 124: please define “AU.”
This has been defined.
- Lines 119, 129, 133, 168, 205: Please change “Nabs” to “sNAbs.”
According to the response to comment 1 and in light of the pVNT results, we did not change Nabs into sNabs.
- Line 155-157: Here the authors discuss the ratios of maternal to fetal antibodies and draw conclusions based on their one sample. It is difficult to draw a conclusion based on an n= 1. Thus explanation should be remove and conclusions included into the next sentence.
We have adapted this section of the manuscript and we have tempered our conclusions.
- Sentence on lines 167-169: This sentence should be rewritten. Try, “It is also interesting to note that total antibodies and sNAbs titers in the maternal blood were at the lower range of our control cohort (Figure 1).”
This sentence has been adapted, as suggested by the reviewer.
Submission Date
23 July 2021
Date of this review
03 Aug 2021 18:00:56
Reference :
- Favresse, J.; Gillot, C.; Di Chiaro, L.; Eucher, C.; Elsen, M.; Van Eeckhoudt, S.; David, C.; Morimont, L.; Dogné, J.-M.; Douxfils, J. Neutralizing Antibodies in COVID-19 Patients and Vaccine Recipients after Two Doses of BNT162b2. Viruses 2021, 13, 1364, doi:10.3390/v13071364.

Reviewer 2 Report
The case-report has been written so well. I have no comment on it.
Author Response
The case-report has been written so well. I have no comment on it.
We thank the reviewer for this very positive feedback.

Reviewer 3 Report
The article title is a case report and literature review. There is no formal literature search with search terms and data bases searched. The article should be shortened to a brief case report note in a couple of paragraphs.
Author Response
The article title is a case report and literature review. There is no formal literature search with search terms and data bases searched. The article should be shortened to a brief case report note in a couple of paragraphs.
We thank the reviewer for this comment, and we have changed the title accordingly. However, we did not shorten the article given the feedback and the comments of the other reviewers and the editor which did not request a reduction of the length of our article.

Reviewer 4 Report
In this article, the authors presented a case study on the successful maternal-fetal transfer of antibodies after vaccination in a pregnant woman at 25 weeks of gestation.
They found that neutralizing antibody levels were about 5 times higher in the umbilical cord than in maternal blood, while the total antibody level only showed a 2-fold increase. They compared their data with sera obtained from 13 non-pregnant women and with literature data for disease-induced or vaccination-induced SARS-CoV-2 maternal immunization, evaluating neonatal protection through transplacental transfer of antibodies.
Although the paper contains data obtained in only one subject, they obtained interesting data on the transplacental transfer of antibodies. I only have two small concerns:
- the study was conducted using BNT162b2 vaccine and there is no reference to whether this has been used on pregnant women before. Please add also the name of company that makes the vaccine
- The caption for table 1 is missing (Table 1. This is a table. Tables should be placed in the main text near to the first time they are cited.)
Author Response
In this article, the authors presented a case study on the successful maternal-fetal transfer of antibodies after vaccination in a pregnant woman at 25 weeks of gestation.
They found that neutralizing antibody levels were about 5 times higher in the umbilical cord than in maternal blood, while the total antibody level only showed a 2-fold increase. They compared their data with sera obtained from 13 non-pregnant women and with literature data for disease-induced or vaccination-induced SARS-CoV-2 maternal immunization, evaluating neonatal protection through transplacental transfer of antibodies.
Although the paper contains data obtained in only one subject, they obtained interesting data on the transplacental transfer of antibodies. I only have two small concerns:
- the study was conducted using BNT162b2 vaccine and there is no reference to whether this has been used on pregnant women before. Please add also the name of company that makes the vaccine
We thank the reviewer for notifying us this oversight. This has been added in the new version of the manuscript.
- The caption for table 1 is missing (Table 1. This is a table. Tables should be placed in the main text near to the first time they are cited.)
We thank the reviewer for having seen this oversight again. This has been completed accordingly.

Round 2
Reviewer 3 Report
The authors have made the changes requested by the three reviewers and have decided to delete literature review from their title and change it to discussion of the literature and have added some citations.